# The Role of Fe, Zn, and Cu in Pregnancy

**DOI:** 10.3390/biom10081176

**Published:** 2020-08-12

**Authors:** Konrad Grzeszczak, Sebastian Kwiatkowski, Danuta Kosik-Bogacka

**Affiliations:** 1Department of Biology and Medical Parasitology, Pomeranian Medical University in Szczecin, Powstańców Wielkopolskich 72, 70-111 Szczecin, Poland; konrad.grzeszczak@pum.edu.pl; 2Department of Obstetrics and Gynecology, Pomeranian Medical University in Szczecin, Powstańców Wielkopolskich 72, 70-111 Szczecin, Poland; kwiatkowskiseba@gmail.com; 3Independent Laboratory of Pharmaceutical Botany, Pomeranian Medical University in Szczecin, Powstańców Wielkopolskich 72, 70-111 Szczecin, Poland

**Keywords:** microelements, iron, copper, zinc, pregnancy

## Abstract

Iron (Fe), copper (Cu), and zinc (Zn) are microelements essential for the proper functioning of living organisms. These elements participatein many processes, including cellular metabolism and antioxidant and anti-inflammatory defenses, and also influence enzyme activity, regulate gene expression, and take part in protein synthesis. Fe, Cu, and Zn have a significant impact on the health of pregnant women and in the development of the fetus, as well as on the health of the newborn. A proper concentration of these elements in the body of women during pregnancy reduces the risk of complications such as anemia, induced hypertension, low birth weight, preeclampsia, and postnatal complications. The interactions between Fe, Cu, and Zn influence their availability due to their similar physicochemical properties. This most often occurs during intestinal absorption, where metal ions compete for binding sites with transport compounds. Additionally, the relationships between these ions have a great influence on the course of reactions in the tissues, as well as on their excretion, which can be stimulated or delayed. This review aims to summarize reports on the influence of Fe, Cu, and Zn on the course of single and multiple pregnancies, and to discuss the interdependencies and mechanisms occurring between Fe, Cu, and Zn.

## 1. Introduction

Macroelements and microelements are essential for the proper functioning of living organisms (Figure 1). They participatein many processes, including cellular metabolism and antioxidant and anti-inflammatory defenses, and also influence enzyme activity, regulate gene expression, and take part in protein synthesis [1]. Provided in the diet, their levels in the human body depend on the geographical location, environmental pollution, gender, and age. A considerable body of research has been devoted to the concentrations of iron (Fe), copper (Cu), and zinc (Zn) in biological materials from women during childbirth, including fetal membrane (FM) and serum (FS), cord blood(CB), serum (CS) and plasma (CP), and maternal blood(MB), serum (MS) and plasma (MP), and placenta (P) (Table 1).

During pregnancy, the diet should meet the needs of the baby, as well as that of the mother, whose health is closely related to the provision of adequate amounts of essential elements including Fe, Cu, and Zn. Their levels before pregnancy can also be of significance [22]. The study by Caan et al. [23] showed that the Special Supplemental Nutrition Program for Women, Infants, and Children (WIC) initiated 5–7 months before pregnancy resulted in an increase in birth weight by 131 g and length by 0.3 cm. Providing the right amount of essential elements with a diet or by supplementation can reduce the risk of fetal malformation and preterm birth [24], including multiple pregnancies—associated with a higher risk of perinatal complications [25]. These complications are likely to cause premature birth, miscarriage, hypotrophy of one or both fetuses, preeclampsia, fetal death, or fetal atrophy syndrome [26].

## 2. Iron (Fe)

Iron (Fe) is a microelement necessary for the proper functioning of living organisms. An adult female body contains about 2 g of Fe (42 mg Fe/kg body weight, b.w.), mainly in the hemoglobin(Hgb;~60%), in ferritin and hemosiderin in the liver, spleen, and bone marrow, and myoglobin and some enzymes, including catalase, peroxidases, and cytochromes [27].

Fe is involved in many metabolic processes, including oxygen respiration, detoxification of reactive oxygen species (ROS), drugs and xenobiotics, and the synthesis and metabolism of various compounds, such as hormones, myelin, neurotransmitters, nucleic acids, and heme. Fe participates in the transport of molecular oxygen from the lungs to all body tissues and cells [28]. This element takes part in erythropoiesis and immune reactions affecting the humoral and cellular immunity of the body [29,30,31,32]. The iron is essential trace elementfor collagen synthesis and the conversion of 25-hydroxyvitamin D into an active form [33]. Fe plays an important role in electron transport [34] and takes part in the regulation of the cell cycle by influencing the expression of certain genes, such as protein kinase C, nitric oxide synthase, and cyclin dependent kinase inhibitor1A(p21) [32,35,36].

Iron is supplied to the human body with the food. A balanced diet for men should contain about 1–2 mg Fe per day [37], while women, due to blood loss during menstruation, should intake 4–5 mg per day [38]. During pregnancy, this level should increase from 4 mg in the 1st trimester to 8 mg in the 2nd trimester, and 15 mg per day in the 3rd trimester [39], due to the increased demand associated with fetal growth, placenta, Hgb formation, and the increase in maternal muscle mass, especially in the 2nd and 3rd trimester of pregnancy [40]. During pregnancy, plasma volume increases by 40–60% and red cell mass by 18–25% [41,42,43]. Hemoglobin concentration decreases to 10.5–11 g/dL and the hematocrit value to 30–32% [42,44]. This process, known as hemodilution, usually intensifies between the 17th and 36th weeks of pregnancy [43]. Additionally, it may result in a slightly decreased vitamin B_12_ concentration (in about 10–28% of women) [45] and a slight decrease in platelet count [46]. It is also worth noting in that during this time (i.e., 17–36 weeks) the number of white blood cells increases by about 20% due to increased hormonal activity (e.g., glucocorticosteroids) and cytokine synthesis (an increase in granulocyte-macrophage colony-stimulating factor) [47]. The World Health Organization (WHO) and the Centers for Disease Control and Prevention (CDC) define anemia in the 1st and 3rd trimesters as Hgb concentrations below 11.0 g/dL and hematocrit below 33%. In the 2nd trimester the values are lower, 10.5 g/dL and 32%, respectively [48,49]. Large amounts of inorganic Fe (non-heme Fe–Fe^3+^) can be found in lentils (8.6 mg/100 g), spinach (3.1 mg/100 g), and broccoli (1.1 mg/100 g) [50]. In contrast, organic Fe (heme Fe–Fe^2+^) is mainly found in meat, such as pork liver (19 mg/100 g) and beef (3.1 mg/100 g) [50]. The assimilability of Fe^3+^ is about 10% and Fe^2+^ up to 50% [51].

Human food most often contains Fe^3+^, reduced by ferrireductase to Fe^2+^ in the intestinal lumen [52]. Fe^2+^ absorption is based on the principle of active transport with a divalent metal transporter (DMT-1) in the apical membrane of the duodenum and the upper part of the small intestine [53,54]. The divalent metal transporter 1 (DMT1) is a non-selective transporter of bivalent metal ions, including iron, zinc, copper, manganese, cobalt, and cadmium, whose transport through the membrane occurs via proton-coupled divalent metal ion transporters [55,56]. Fe levels may also be influenced by the expression of intestinal copper transporting P-type ATPases copper-transporting ATPase α (ATP7A), which indirecly impairs Fe absorption by affecting the expression of Fe transporters [57]. Some researchers suggest a separate mechanism for the absorption of heme Fe which involves heme carrier protein (HCP1), where Fe^2+^ is transported through the apical membrane of the small intestine to enter the enterocytes, and then binds to apoferritin to form ferritin, a compound used to store Fe^2+^ [58,59]. However, not all researchers agree that HCP1 plays this role [60]. For example, a study by Qiu et al. [61] confirms the thesis that patients with a mutation in the gene encoding HCP1/PCFT have a folate deficiency, with Fe metabolism intact. This shows that mechanism of heme absorption via the intestines is still undefined [60]. The surplus Fe^2+^ in the enterocytes is transported to the bloodstream by ferroportin (IREG1) and reoxidized to Fe^3+^ by hephaestin and ceruloplasmin, then immediately bound to the major iron-binding plasma proteinapotransferrin to form transferrin [62,63]. In a study on zebrafish (*Danio rerio*), Donovan [64] demonstrated that IREG-1 on the surface of placental syncytiotrophoblasts is involved in the transport of Fe from the mother to the fetus. The main protein transporting Fe^3+^ to cells is transferrin [65]. Equally important in Fe turnover is lactoferrin found in breast milk—a source of Fe^3+^ for newborns and infants [66]. The assimilability of Fe from lactoferrin is about 50%, while the assimilability of Fe from cow’s milk is only about 5% [29]. Another important source of Fe in the body are macrophages, which recover Fe^2+^ from erythrocytes [67]. Macrophagesdegrade Hgb, resulting in the release of Fe^2+^, which is then transported by the transmembrane ferroportin transporter (FPN1) and oxidized by ceruloplasmin to Fe3^+^, to finally bind to transferrin [68]. The whole process is controlled by hepcidin, a peptide synthesized in hepatocytes, which regulates both the amount of Fe absorbed by enterocytes and Fe^2+^ released by macrophages [69,70].

The bioavailability of Fe in the bloodstream affects the absorption of Fe in the placenta and increases with gestational age [71]. It is likely that the Fe balance between the woman and the fetus is controlled by hepcidin [72]. Thanks to transferrin receptor 1 (TFR 1), Fe combined with transferrin can be transferred to placental syncytiotrophoblasts from the bloodstream [73,74,75]. Bastin et al. [76] showed the expression of TFR 1 on the apical maternal-facing membrane and, additionally, the expression of ferroportin (FPN) which was localized on the basal fetal-facing membrane. This is the direction that suggests the transport of Fe from mother to fetus. Moreover, the authors noted that ferritin was strongly expressed in the stroma fetal tissue, which suggests iron storage. In the next stage, Fe^3+^ is reduced to Fe^2+^ by ferrireductases STEAP 3 and STEAP 4 [77]. Subsequently, the transport of Fe from the endosome, through the basale membrane to the fetus, takes place via DMT-1 [78] and/or ZIP 8 and ZIP 14 [79]. However, Gunshin et al. [80] challenged the theory of DMT-1 involvement in the transport of Fe in the placenta. In their study, the mouse gene DMT-1 was inactivated, but this did not prevent the birth of live offspring and confirmed the effective transfer of Fe through the placenta. Therefore, ZIP 8 and 14 are more important than DMT-1, but current reports are not conclusive for humans in this case [79,81]. The second interesting source of Fe supply to the fetus may be ferritin, heme, and non-transferrin-bound iron (NTBI), but unfortunately at this point it has not been confirmed by research. The entire mechanism described in this paragraph is not fully understood and we need considerably more research on how exactly the placenta delivers Fe to the fetus [82,83].

Small amounts of Fe are excreted from the body with urine, saliva, sweat, and also as a result of physiological blood loss in the digestive tract [84,85,86]. In women, epithelial exfoliation and blood loss during menstruation cause a loss of about 1 mg of Fe [38]. To compensate for this loss, a similar amount of Fe is absorbed from the gastrointestinal tract [87]. During pregnancy, intestinal absorption of Fe in the 2nd and 3rd trimester increases by about 3 mg compared to the physiological state [88]. The high loss of Fe from the body is compensated by reserves stored in hepatocytes or macrophages [89]. The correct Fe concentration in blood should be 80–130 μg/L and remains in dynamic equilibrium with the concentration of this element in erythrocytes, bone marrow, and in its free form [27].

Fe deficiency may occur as a result of a low Fe supply [90], increased blood loss [91], intravascular hemolysis with hemoglobinuria occurring in malaria [92], *Ancylostoma duodenale* [93], Fe absorption disorders from the gastrointestinal tract in the course of e.g., *Helicobacter pylori* infections, inflammatory bowel diseases, coeliac disease [94], congenital or acquired transferrin deficiency [95], cancer [96], and increased demand for Fe, e.g., during pregnancy [97]. The risk of Fe deficiency is high among women during pregnancy and lactation, children and adolescents during intensive growth, vegetarians, and in the elderly [98]. In addition, poisoning or oversupply of certain heavy metals, including lead, manganese or cobalt may also cause Fe deficiency. [99]. Fe homeostasis is particularly vulnerable to Pb due to its ability to bind to DMT1, which results in the secondary inhibition of Fe absorption [100].

In a study on laboratory animals (C57BL/6 mice), Hubbard et al. [101] found that removal of Fe from the diet reduced the birth weight of the offspring and increased the risk of stillbirth. Woodman et al. [102], in a study on Sprague-Dawley rats, evaluated mitochondrial function and ROS generation in animals totally or partially deprived of Fe in their diet. The authors showed an increase in the number of mitochondria in male kidneys. Additionally, they found an increase in cytosolic superoxide inmale kidneys and liver and in female kidneys. The results suggested that male fetuses were more susceptible to mitochondrial disorders and oxidative stress than female fetuses.

At the beginning of pregnancy, a woman’s body uses tissue reserves of Fe (e.g., from hepatocytes) [103]. In laboratory tests, the only indicator is a decrease in serum ferritin concentration at normal serum Fe levels [104]. Then, as a result of increased hematopoiesis and the development of the fetus and placenta, the Fe reserves in the mother’s body are exhausted [105]. This results in hypochromic microcytic anemia, especially in the 2nd and 3rd trimester, which may give adverse symptoms of organ and tissue dysfunction [106]. Premature birth or a low weight birth may occur [107], as well as myocardial hypoxia (e.g., tachycardia) [108], cerebral hypoxia (e.g., weakness, drowsiness, headaches, and dizziness) [109,110,111], immunosuppression [111], and appetite disorders (e.g., eating chalk) [112]. The prevalence of anemia in pregnant women ranges from 17% to 31% in Europe and North America, 53–61% in Africa, and 44–53% in Southeast Asia [113]. Milman et al. [114] found that the prevalence of iron deficiency (ID) and iron deficiency anemia (IDA) in pregnant women from Europe is 10–32% and 2–5%, respectively, and the lack of Fe supplementation during pregnancy increased the prevalence. Only 20–35% of women of childbearing age did not require additional Fe supplementation.

Insufficient supply of Fe can cause disorders in oxygen transport and consequently lead to anemia [100]. It is particularly dangerous for women of reproductive age due to blood loss during menstruation and also during pregnancy [115]. Fe deficiency may cause preeclampsia and premature fetal membrane rupture in pregnant women [116], and also lead to a decrease in the child’s vital signs [117,118]. The meta-analysis of Figueiredo et al. [119] proved that maternal anaemia was associated with a higher risk of low birth weight. Pregnant women, especially those with multiple pregnancies, should monitor Fe levels from the beginning of the pregnancy. Low Fe concentration at the beginning of pregnancy significantly correlates with the occurrence of anemia in the last trimester of pregnancy [120]. Therefore, it is recommended to supplement pregnant women’s daily diet with 27 mg of Fe per day, and with as much as 60 mg if they are diagnosed with anemia [121,122].

Fe-containing preparations have been found to have a positive effect on the course of pregnancy and in the perinatal period. The supplementation of pregnant women with this micronutrient may increase the bodyweight of newborns by 200 g on average [123]. On the other hand, excessive Fe supplementation may lead to an increase in ROS formation, which in turn leads to tissue and organ damage [124]. Peña-Rosas et al. [125] evaluated the effect of Fe supplementation in pregnant and perinatal women and noted that daily Fe supplementation had a beneficial effect on the course of pregnancy, reducing the risk of low birth weight of the newborn, and preventing the occurrence of anemia in women. Abioye et al. [126] studied the effect of taking Fe supplementation in a group of pregnant women with a deficiency of this element, assessing the value of hematological parameters in patients before and after birth. The authors demonstrated a significant increase in the levels of ferritin, hepcidin, and Hgb, and a decreased level of soluble transferrin receptor (sTfR). The increase in Hgb concentration in the mothers also resulted in a reduced risk of newborn death. Similar results concerning an increase in Hgb concentration and a decrease of anemia in women were described by Zhao et al. [127]. Peña-Rosas et al. [128] found that pregnant women taking daily preparations containing Fe were less at risk of adverse events and anemia during pregnancy than women who supplemented this element irregularly. In contrast to other researchers, Ziaeii et al. [129] showed that there was no significant relationship between the dosage of Fe-containing supplements and ferritin concentration in pregnant women. Ali et al. [130] also studied the effects of different levels of Fe supplementation in women pregnant with twins from 12 to 36 weeks of pregnancy. One group of women received 27 mg of Fe daily and the other group 54 mg. In both groups, Hgb and hematocrit (Hct) levels were normal throughout the study period, and in only one group was the mean ferritin level higher. The authors concluded that a daily intake of 27 mg Fe is as effective as a higher dose of Fe. The higher level of in Fe supply did not affect Hgb and Hct levels, but it did increase the incidence of side effects (nausea or vomiting). In the study by Shinar et al. [131] on 172 women pregnant with twins with IDA, divided into two groups receiving 34 and 68 mg of Fe sulfate, showed that higher doses of Fe caused higher Hgb values in the examined women.

Acute poisoning was observed as a result of an overdose of Fe from a pharmaceutical preparation [132]. If the transferrin potential is exhausted or exceeds 85% in the blood, non-transferrin-related Fe (NTBI) may occur, increasing the risk of intraocular and cardiac Fe accumulation [133]. NTBI initially appears in the bloodstream and then is transported to the interstitial cells via a transferrin-independent mechanism [134]. It was initially assumed that DMT1 transporter is responsible for the transfer of NTBI to the liver and then its removal, but a study on DMT1-negative mice showed they were not protected against the accumulation of Fe in the liver. Therefore, another mechanism was proposed, consisting in the ability of Zip14 to uptake NTBI [135]. Two types of hemochromatosis have been identified: primary (hereditary) and secondary (acquired, due to excessive accumulation of Fe in the body) [136]. As a result of excessive Fe accumulation in organs, such as the heart, liver, and endocrine glands, their function is impaired, producing cardiomyopathy, cirrhosis, or insulin-dependent diabetes [136,137]. In case of pregnancy, the liver metabolism error described above had a negative effect on the fetus’s Fe balance, which was emphasized in the study by Silver et al. [138]. The placenta acquired excessive amounts of Fe and passed them to the fetus, although the authors could not identify any underlying mechanism of this effect, indicating the necessity for further research on the metabolism and kinetics of Fe transport to the fetus.

A high accumulation of Fe results in the reduced absorption of other essential elements including zinc, copper, molybdenum, chromium, manganese, and magnesium, and the production of ROS [139]. Oxidative stress is particularly dangerous for pregnant women, who may suffer damage to the placental tissue and consequently a premature birth [124]. Oxidative stress leads to faster aging of the cells, which gives a signal to the uterus to contract, resulting in premature birth. This process can also lead to the rupture of the fetal membrane [140].

## 3. Copper (Cu)

Copper is an essential micronutrient. It has been estimated that the amount of this element in an adult human body ranges from 50 to 120 mg [141]. The highest concentrations of Cu were found in the brain and the liver [142]. Cu is involved in the formation and metabolism of bone tissue and participation in oxidation–reduction reactions as a coenzyme, a regulator of Fe metabolism and transport, as well as collagen metabolism [122,143]. Cu participates in the metabolism of fatty acids, in RNA synthesis, supports the absorption of Fe in the gastrointestinal tract, and participates in the synthesis of myelin [144,145]. Cu also participates in the synthesis of melanin and—as a component of tyrosinase—is involved in the conversion of tyrosine to melanin [27].

The copper is delivered with the food. The greatest source of this element is oysters (44,996 µg/100 g), beef liver (6434 µg/100 g), cocoa (5000 µg/100 g), sunflower seeds (1770 µg/100 g) [146]. The daily supply of Cu should be 0.9–1.3 mg per day, although it is mostly consumed in larger amounts—about 2 mg per day [147]. During pregnancy, the demand for Cu increases to 1 mg/day [122]. Cu homeostasis is determined by the balance between intestinal absorption and biliary secretion [148]. About 50–70% of Cu from food is absorbed in the duodenum and upper sections of the small intestine, and small amounts in the stomach [143,149]. In the stomach, Cu^2+^ is reduced to Cu^1+^ and then absorbed into the intestinal endothelium [150]. Uptake, intracellular transport, and removal of excess Cu is strictly regulated and involves specific proteins. Copper is transported to cells via two transmembrane proteins: high affinity copper uptake protein 1 (CTR1) and DMT1 [151,152,153]. CTR1 is responsible for the transport of 80% of Cu and other metals to cells [154,155]. It is believed that the *Ctr1* mutation may lead to the abnormal functioning of intracellular signaling pathways during embryonic development. Kuo et al. [156] found early embryonic lethality associated with a *Ctr1* null mutation in mice. In cytoplasm, Cu is complexed mainly with metallothionein (MT). Copper ions serve as cofactors of certain enzymes to which they bind via cytoplasmic copper chaperones. Copper chaperone for superoxide dismutase 1 (CSS) and cyclooxygenase (COX) and synthesis of cytochrome c oxidase (SCO) proteins are responsible for the transport of Cu to superoxide dismutase type 1 (SOD1) and cytochrome c oxidase (CCO), respectively. Antioxidant protein 1 (ATOX1) transports Cu ions to copper transporting P-type ATPases ATP7A and copper-transporting ATPase β (ATP7B), which regulate the concentration of Cu ions in the cell and mediate the incorporation of cations of this element into enzymatic proteins [157]. The lack of ATOX1 results in the accumulation of Cu and inhibition of its secretion. The role of six-transmembrane epithelial antigen of the prostate (STEAP) and duodenal cytochrome b (Dcytb) in Cu metabolism is poorly understood [153]. In the blood, Cu occurs in complexes with histidine, threonine, and glutamic acids transported to the liver, kidney, intestines, and other tissues [143,158]. The main organ responsible for Cu metabolism is the liver, which accumulates this element in the prenatal and postnatal period, as well as synthesizing ceruloplasmin and producing bile, which has a high Cu concentration [122,143,144,145,148,149,150,158,159,160,161,162].

Cu is excreted from the body with urine and feces. Only 2% of Cu^1+^ is excreted through the kidneys because in proximal tubules Cu is mostly reabsorbed by ATP7A proteins and then returned to the bloodstream [163]. In the liver, the excess Cu is transported via ATP7B and bound with substances contained in the bile and passed to the digestive tract where it is removed with the feces [164,165]. In this way, more than 90% of the excess Cu is excreted, and its incorporation into ceruloplasmin prevents reabsorption from the intestines.

The process of transporting Cu to the placenta from the maternal circulation is probably caused by the copper transport protein 1 (CTR1), as shown by Lee et al. [166], who deprived the mouse of CTR1 and observed teratogenicity or fetal mortality in the uterus. When Cu is delivered to the placenta, the element is attached to the copper-transporting P-type ATPases (ATP7A or ATP7B) [167]. The ATPases are coordinated with each other as ATP7A is located on the basolateral membrane and is responsible for the excretion of Cu into the fetal circulation. In contrast, ATP7B is in a perinuclear compartment or the microvillar membrane, releasing excess Cu into maternal circulation [168]. In contrast, McArdle et al. [169] do not agree with this model of transporting Cu in the placenta and believe that this mechanism should be further investigated.

Cu deficiency is rare due to the high availability of this element in food [170]. Cu deficiency may lead to pancytopenia and the occurrence of hypodermic anemia which does not respond to the administration of Fe [171,172], loss of appetite [173], damage to internal organs [174,175], bone deformities [176], reduced reproductive capacity [177], myocardial fibrosis [178,179], and chronic debilitating diarrhea [180]. It may lead to neurological disorders, and hair depigmentation [181,182]. A deficiency of this micronutrient during pregnancy may lead to oxidative stress, which often results in reduced fetal growth [122]. Cu has an important role in the production of collagen and elastin, and an insufficient amount of this element can lead to a reduction in the tensile strength of the fetal membrane, resulting in its interruption and premature birth [183]. On the other hand, excessive administration of Cu causes vomiting, diarrhea, as well as liver necrosis, acute kidney damage, and—ultimately—death [184,185].

Several diseases related to abnormal Cu metabolism have been described, especially those caused by Cu deficiency. An example is Menkes disease which is caused by the absence of or defecst in two Cu-transporting ATPases encoded by the *ATP7A* gene, which affects the absorption of Cu ions in the gastrointestinal tract [186]. This condition, found in one per 300,000 live birthsin the European population [186], is due to the mutation of the *ATP7A* gene on the chromosome 3 that causes X-linked recessive disorders [187]. It affects almost exclusively men, with women considered to be only carriers of the mutation, except a few isolated female cases reported in literature [188]. Menkes disease is characterized by significant physical and mental impairment of patients and death in early childhood [187]. In rare cases, the activity of ATP7A protein is partially preserved and the symptoms milder (occipital Horn syndrome), with patients living as long as 50 [171]. The mutation in the *ATP7A* gene does not necessarily lead to Menkes disease; it has also been described in patients suffering from hereditary motor neuropathy which results in the weakness of the upper limbs that progresses with age [189]. Another disease leading to the disorder of Cu metabolism is Huppke-Brendel syndrome, resulting from the mutation of the *SLC33A1* gene encoding the AT-1 protein, first described in 2012 [190]. People with this syndrome have very low levels of Cu and ceruloplasmin, and often do not reach the age of 6. The symptoms of this disease include psychomotor disability, hypoplasia, and hypomyelination, probably due to the lack of ATP7A protein acetylation. Another disease affecting Cu homeostasis is MEDNIK syndrome (mental retardation, enteropathy, deafness, ichthyosis, keratoderma), caused by the mutation of the *AP1S1* gene located on the long arm of chromosome 7 at position q22 [191]. The patients suffer from mental and motor impairment and deafness; other characteristic symptoms include dermatological changes including excessive keratosis and exfoliation of the epidermis with features of fish scale (lat. ichthyosis) and erythroderma.

Kashaninet al. [192] studied the effects of Cu on pregnancy. They showed that women who had been supplemented with 1000 mg of Cu daily from the 17th week of pregnancy experienced reduced depression symptoms in the 2nd and 3rd trimesters of pregnancy compared to the control group. Jankowski et al. [193] demonstrated in an in vitro study on rat embryos that Cu is essential for early embryonic development. The fetus is sensitive to a lack of Cu, which may lead to irreversible changes in the nervous system [194,195]. Meta-analysis by Lewandowska et al. [196] showed that reduced serum Cu levels at the beginning of pregnancy were associated with a higher risk of pregnancy-induced hypertension. Prohaska and Brokate [197] showed that a correct amount of Cu in the body has a protective effect during pregnancy. Vukelići et al. [198] found that in women with normal fetal development, Cu levels were significantly higher than in women with fetal developmental disorders. Another study also showed that Cu protects against spontaneous miscarriages [16]. However, elevated Cu levels in the first trimester of pregnancy may lead to later pregnancy complications, including spontaneous miscarriages [199]. Li et al. [200] concluded that too much Cu can increase the incidence of gestational diabetes, which—if untreated—can lead to macrosomia, intrauterine hypotrophy, birth defects, and miscarriages.

An excess of Cu is harmful but rare [201]. It results in the formation of free radicals that damage cell membranes and proteins in the body (the so-named Fenton Reaction) [202]. Excess Cu has been observed in the course of Wilson’s disease (WD) as a result of abnormal cellular transport [203]. In pregnant women with WD, it is important to take early treatment to prevent spontaneous miscarriages [204,205]. In pregnant women, Cu levels may be elevated due to increased levels of Cu-bearing proteins [206] and may cause premature births and low birth weight compared to pregnancies with normal Cu levels [122].

## 4. Zinc (Zn)

As an essential trace element (micronutrient), Zn has several important roles in human bodies, and after Fe, it is the second most abundant trace element [207]. The total Zn content in the human body is 2–4 g, with a plasma concentration of 12–16 μM/L [208,209,210]. In cells, Zn is present as a divalent ion (Zn^2+^) [211]. The highest Zn concentrations have been found in skeletal muscle (60%), bones (30%), liver (5%), and skin (5%) [212].

Zn is a co-factor of more than 300 enzymes that regulate a variety of cellular processes and cellular signaling [213]. It is responsible for the DNA-binding ability of many transcription factors through the unique ability to form molecules known as Zn finger (Znf) proteins [214]. Zn is essential for cell division, differentiation, and the development of organs, such as the kidneys and heart [215]. It is required for normal testicular development [216]. Zn regulates reproduction, fetal development, membrane stability, digestion, wound healing, and homeostasis of the central nervous system [217,218,219]. As a cofactor for the enzymes, Zn participates in bone mineralization and collagen structure development of the bone [220]. Zn influences the function of many hormones, including growth hormone, insulin, testosterone, and gonadotropins [221]. It plays a role in the synthesis, storage, and secretion of insulin. This element is also a component of thymulin, a thymus hormone, essential for maturation and differentiation of T cells [222,223]. Zn participates in the elimination of free peroxide radicals, as part of peroxide dismutase (Zn-SOD) [224]. It inhibits the oxidation of unsaturated fatty acids [225], regulates the concentration of vitamin A in plasma [226], and acts as an antagonist of cadmium and lead [227].

Recommended daily intake of Zn depends on several factors such as age, sex, weight, and phytate content of the diet [228]. Food products containing the most Zn (per 100 g) are calf liver (8.4 mg), pumpkin (7.5 mg), pork liver (4.5 mg), and white beans (3.8 mg) [159]. Zn is best assimilated from meat products (red meat, fish) than from plant products [229]. The degree of bioavailability of Zn from food depends on the concentration of Zn in the body, stress [230,231], functional state of the kidneys [232] and/or liver [233], pH of the nutrient content [234]. The presence of citric acid [235], a high intake of animal proteins [236], the presence of highly processed products and lactose [237] have a positive influence on the bioavailability of Zn. Negative effects on Zn absorption are by heavy metals (cadmium and mercury), calcium, Fe, Cu, oxalates, phosphates, alcohol, folates, and phytates, alkaline drugs, contraceptives, and diuretics [238,239,240,241].

Daily Zn supply according to WHO should be 6.5 mg/day for women and 9.4 mg/day for men (both over 19 years of age) [242]. Due to hemodilution and decreased albumin levels, Zn serum concentration declines during pregnancy [243]. Since intestinal absorption is not increased during pregnancy, an additional Zn requirement for fetal and placental tissues must be covered by increased intake and from maternal tissues. Therfore, daily requirements for Zn during pregnancy range from 7.3 to 13.3 mg [244]. Zn enters the human body through the digestive and respiratory systems, and skin. Zn is absorbed by passive diffusion in the intestine by specific zinc transporting proteins [245]. The most important Zn transporters include Zrt-/Irt-like protein (ZIP) zinc transporters and ZnTs (zinc transporters) [246]. The zinc transporter protein-1 (ZnT1)–ZnT10 which decrease cytoplasmic zinc level by zinc export, and Zip (Zip 1–14) which increase cytoplasmic zinc level by zinc import [246]. The ZIP4 transporter can be found in the entire digestive tract and is responsible for Zn transport from the apical membrane into enterocytes [247]. Additionally, similar to Cu, DMT1 transporter may help in the absorption of Zn, by transporting Zn from the intestinal lumen into the enterocytes [248].

The average absorption of Zn from food is 33% [249]. Zn^2+^ ions in the plasma are transported to the cells mainly as complexes with albumin (84%), alpha-2-macroglobulin (15%), and amino acids (1%) [250,251]. They also form complexes with cysteine and histidine [252,253]. Metallothioneins (MT) play an important role in cellular transport [254]. These low-molecular proteins, rich in cysteine residues, are located in the extra- and intracellular environment [255]. Their intracellular pool acts as a reservoir of important heavy metals, including Zn, and participates in the detoxification of ROS, nitrogen, excess heavy metals, and organic compounds [256,257,258,259]. Their extracellular pool is responsible for the transport of Zn^2+^ (and other heavy metals) and organic compounds, and is also a free radical scavenger [256]. One MT molecule is capable of binding seven Zn^2+^ ions and up to 12 monovalent Cu ions [260,261]. The intra- and extracellular Zn concentration is controlled by two families of transporters: ZnT (SLC30-Soluble Carrier 30) and ZIP (SLC39A1-A14) [262,263]. ZIP proteins are responsible for the transport of Zn^2+^ to the cell cytoplasm [264]. ZnT proteins originate from a large family of Cation Diffusion Facilitator (CDF) proteins and are responsible for transporting Zn ions from the cytoplasm of the cell to cell organelles or outside the cell [265]. The main organ responsible for the metabolism and systemic Zn homeostasis is the liver [266]. Zn is excreted mainly in feces (90%) and the rest of Zn is excreted in the urine, sweat, and saliva, and incorporated into hair [267].

The transport of Zn to the fetus is fully dependent on its concentration in the mother’s blood [268]. Its low intake will reduce the expression of ZnT, resulting in less Zn being transported in the placenta, protecting the mother against Zn homeostasis disorder. Zinc released from enterocytes into the mother’s blood through ZnT1 reaches placental syncytiotrophoblasts using ZIP proteins [269]. In the next step Zn can be intercepted by transporters including ZIP 14 and ZIP 8, ZnT2 and DMT-1 or bind to MT and in this form delivered to the fetus [81,270,271]. This mechanism requires further investigation.

A Zn deficiency caused by inadequate dietary intake is a common phenomenon and affects up to two billion people in the world, mainly in developing countries [272,273]. Marginal Zn deficiency is estimated to affect 82% of pregnant women worldwide [274]. Zn deficiency can be classified as primary or secondary. Primary Zn deficiency may lead to growth retardation, puberty delay, immune deficiency, and the impairment of cognitive function, taste, and smell [275]. Major manifestations of Zn deficiency include growth retardation, testicular and ovarian dysfunction, immune dysfunctions, and cognitive impairment [276]. Maternal Zn deficiency during pregnancy is linked with adverse pregnancy outcomes including abortion, preterm delivery, stillbirth, and fetal neural tube defects [277,278,279]. Secondary Zn deficiency is caused by gastrointestinal diseases, including malabsorption syndrome and Crohn’s disease [280]. It may also be caused by heavy hemorrhage, cirrhosis, kidney disease, and alcoholism [281,282,283].

Zn takes part in the embryogenesis and formation of the fetus, and so its low level may result in impaired development and affect the final phenotype of the newborn’s organs [284,285]. Additionally, during pregnancy, a deficiency may contribute to preterm delivery, pregnancy-induced hypertension, low birth weight, and preeclampsia [286]. Zn deficiency may lead to faulty estrogen function, resulting in uterine spasms, cervical dilation, and amniotic fluid integrity [287]. Zn deficiency is also involved in the synthesis of prostaglandins, as well as collagen synthesis and degradation, and so its absence may result in premature fetal membrane rupture [280]. In laboratory animal studies, Zn deficiency during the early stages of pregnancy is associated with reduced fertility [288] and the increased incidence of nervous system congenital malformations [289]. Zn deficiency in later stages of a pregnancy negatively affects neuronal growth and brain function and synaptogenesis and may be associated postnatally with impaired brain function and behavioral abnormalities [290].

Scientific literature data on the influence of Zn supplementation on the course of pregnancy and fetal development are divergent. Nossier et al. [291] concluded that Zn intake minimizes the risk of premature birth in women and infections in newborns. A double-blind and randomized controlled study shows that Zn supplementation during pregnancy improves birth length after adjusting for maternal height and pre-pregnancy weight [292]. Wang et al. [293] in a cohort study on 3187 pregnant women from China, concluded that Zn deficiency in the diet of pregnant women increases the risks of low birth weight and small gestational size. Merialdi et al. [294] noted that prenatal supplementation of Zn-deficient mothers may be beneficial to fetal neurobehavioral development. In an earlier study, Merialdi et al. [295] concluded that the fetuses of mothers who received Zn supplementation showed fewer episodes of minimal fetal heart rate variability, increased fetal heart rate range, an increased number of accelerations, an increased number of movement bouts, an increased amount of time spent moving, and an increased number of large movements. Further, Merialdi et al. [296], studying the effect of Zn supplementation of pregnant women on fetal bone growth found that supplementing Zn-deficient mothers with 25 mg Zn/day caused a higher fetal femur diaphysis length. This was caused by this microelement stimulating osteoblast production and inhibiting osteoclast activity.

In a study in laboratory animals (C57B1-6J mice) Wilson et al. [297] proved that the lack of Zn in the diet negatively affected the growth and development of the offspring in the uterus mainly due to impaired placenta development. Additionally, they noted that a lack of Zn affected the capacity of the heart and organ perfusion. Correct blood pressure in the pregnant woman is essential to properly maintain the growth of the fetus and provide appropriate nutrients [298]. Many studies have shown a slight effect of Zn supplementation or lack of it on maternal and fetal bodies [299,300,301]. Meta-analysis by Chaffee and King [302] concluded that the additional supply of Zn does not affect fetal morphometric results. Discrepancies in study results may have been caused by a different study area. In developing countries, shortages of Zn occur quite frequently, compared to developed countries, where such a phenomenon is probably related to low social status, as confirmed by Mori et al. [303]. Finally, King [304] concluded that Zn supplementation is essential to counteract the effects of smoking, alcohol abuse, infections, injuries, and impaired gastrointestinal function.

There are not many reports on Zn supplementation in women with multiple pregnancies. Goodnight et al. (2009) concluded that in this group additional supplementation could bring positive benefits [305]. Mahomed et al. [306] noted that 15–44 mg of Zn supplementation per day was associated with a 14% reduction in preterm birth. Campbell et al. [307] noted the Zn concentration in plasma in twin pregnancies was similar to that in single pregnancies. It may suggest that the body regulates Zn deficiency in case of multiple pregnancies without additional supplementation.

High doses (100–300 mg/d) of Zn disturb many biochemical processes [308]. Excess Zn can manifest itself through nausea, vomiting, abdominal cramps, and diarrhea [309]. According to WHO [310], excess Zn during embryogenesis can be teratogenic or ultimately fatal.

## 5. Fe vs. Cu

Dietary Fe and Cu are absorbed in the proximal small intestine [311]. These elements must be reduced before uptake into enterocytes, then both metals are oxidized after export into the interstitial fluids [312]. Chen et al. [313] showed that Cu-deficient C57BL/6J mice had significantly lower hephaestin and ceruloplasmin (Cp) ferroxidase activity. Cu facilitates the absorption and utilization of Fe [314], which takes part in red cell regeneration [315] and bone formation [316].

Ha et al. [312] proved that with a high supply of Fe, there is a lower Cu concentration in diseased tissue, and serum Cu activity decreases. Ha et al. [317] also analyzed the mechanism responsible for the impairment of Cu absorption by feeding mice with a feed containing an appropriate amount of Fe and low/suitable/high Cu levels. The mice showed no disruption in Cu absorption and utilization in rodents. However, an increased supply of Fe did impair the utilization of Cu in the body, although the absorption of Cu was normal. The authors investigated the locations of absorbed Cu with a radioactive marker, but could not identify most of the absorbed Cu in any of the tested tissues. They concluded that additional experiments should be performed to investigate the mechanism of distribution, storage, and bioavailability of Cu under elevated condition of Fe. The Cu activity affects the transport of Fe mainly during blood loss. Hephaestin (protein homolog of ceruloplasmin) is not able to make up for this deficiency on its own and the body replaces it with Cp [318]. Additionally, the reduced Cp activity prevents the release of stored Fe from the spleen and liver macrophages [319]. This is probably caused by the reduction or inhibition of Fe oxidation via Cp [320].

Harris [321] confirmed the role of Cp in the metabolism and release of tissue Fe. This study was conducted on a ceruloplasmin genetic defectin proteinsynthesis in patients who were diagnosed with excessive accumulation of Fe in the brain and pancreas. This resulted in retinal degeneration and diabetes. The patients were given Cu intravenously which resulted in increased serum Fe concentration. These results confirm a significant influence of Cu on the release of tissue Fe. Ha et al. [322] showed that Cu metabolism is impaired when dietary Fe intake is >10 times higher than the normal Fe supply established by WHO. Significantly elevated levels of Fe may result in anemia, likely due to a lack of Cu in the diet [323]. This is because Cu is a component of cytochrome oxidase, necessary for oxidative phosphorylation, which allows the incorporation of Fe into heme. Additionally, the lack of Cu may shorten the erythrocyte life span [324]. This may be due to the inhibition of superoxide dismutase, which contains Cu, leading to cell membrane damage by ROS. In addition, there is a decrease in Cu activity, which inhibits peroxide formation [325]. Hepatic Fe overload also causes liver damage, causing hypertriglyceridemia and hypercholesterolemia [326] and elevated Fe levels may lead to fibrosis or cirrhosis of the liver, via a decrease in superoxide dismutase activity [327]. On the other hand, when the supply of Cu is increased, the absorption of Fe may be impaired due to the affinity of Cu to transferrin [328]. However, Cu overload is very rare because there is a difference between Fe and Cu in the final metabolism of both elements. Cu is removed from the body with bile, and there is no mechanism of removing excess Fe.

Sebio et al. [329] concluded that elevated concentrations of Fe or Cu in the body cause brain damage due to oxidative stress. Lan et al. [330] found that the accumulation of Fe, Cu, and cobalt in the brain causes a decrease in glutathione, which is responsible for oxidative protection of neurons. In rats, Fe supplementation lowers the level and absorption of Cu. In patients who received 3 mg of Cu from 100 or 400 mg of Fe in the form of Fe gluconate, no effect of Fe on Cu absorption has been demonstrated [331].

The study by Christian et al. [332] on 235 pregnant women showed no negative effect between Zn and Fe when these elements were administered at a 2:1 ratio. The additional effect of supplementation with these elements was shown by Saak [333] carried out in Ghana on 354 pregnant women given a dose of 40 mg of Fe and Zn—an increase in HgB and ferritin values was reported. Although the results of that study suggest a beneficial effect of such a supplementation model, it was effective only for women with Fe deficiency. This was also confirmed in the subsequent study by Saaka et al. [334]. Harvey et al. [335] estimated that even taking 100 mg of Fe with a meal does not affect Zn metabolism. The authors suggest that Zn status is maintained after the effective absorption of Zn from food meals. In the opposite situation, when Zn was supplemented in a diet, it also did not affect the balance of Fe [336]. However, in the study by O’Brien et al. [337] daily Fe supplementation (60 mg) significantly reduced the absorption of Zn and only the inclusion of 15 mg Zn supplementation in the diet of these women reduced this adverse effect of Fe. The authors also noted the need for additional studies in this direction and the establishment of the mechanism that causes such interaction.

## 6. Fe vs. Zn

Zn and Fe interact competitively during intestinal absorption [338]. In a study on animals, Bodiga and Krishnapillai [339] found that the interactions between Fe and Zn during absorption in Fe- and Zn-deficient rats are mutually antagonistic. The competition of Fe and Zn for metaltransporter1 (DMT-1) at the site of absorption results in reduced uptake of these elements during concurrent administration [340]. This was confirmed in a study by Espinoza et al. [56] in which DMT-1 was involved in the active transportof Fe, Cu, and Zn, although Zn showed a different relative capacity. Kordas and Stoltzfus [341] showed that althoughFedoes seem to reduce the absorption ofZn, DMT-1 is an unlikely site for this absorptive antagonism because Znis not transported by DMT-1. Zrt- and Irt-like protein-14 (Zip14) is a transmembrane metal iron transporter that is abundantly expressed in the liver, heart, and pancreas [342].

It was found that a 1:1 mass ratio of Fe/Zn in the diet causes a slight inhibition of Zn absorption, whereas when the Fe/Zn ratio is higher, 2:1 or 3:1, the absorption of Zn is limited. It has been noted that the intake of heme Fe together with inorganic Zn in the appropriate ratio does not impair the absorption of Zn. Organic Zn does not affect the absorption of Fe [343,344].

Pregnant women have an increased demand for Zn and Fe [345]. The administration of Fe at more than 100 mg/day to pregnant women resulted in a reduction in serum Zn compared to the women who were given less than 100 mg of Fe [346]. Women in the first trimester of pregnancy who additionally received multivitamin preparations containing 60–65 mg of Fe showed a reduction in Zn absorption compared to women receiving less than 30 mg Fe/day [347]. Ziaei et al. [348] estimated that HgB greater than 13.2 g/dL in pregnant women reduces serum Zn levels. Andersen et al. [349] suggested that Cu deficiency has not only a direct effect on the concentration of Fe, but also an indirect effect through regulation of Fe transporters, which, inter alia, affects the delivery of Fe to the fetus. On the other hand, Gambling et al. [350] suggests that there is no connection between Fe and Cu in their research. Additionally, Shidfar et al. [351] reports that Fe supplementation does not have a significant effect on Cu levels in the body.

Donangelo et al. [352] examined adult women (*n* = 23; aged 20–28 years), non-anemic but with low Festores. The authors investigated the effect of Fesupplementation (100 mg Fe/day) or Znsupplementation (22 mg Zn/day) for 6 weeks. They found that theuse ofFe supplements in those womenimprovedFeindices with no effect onZnstatus. However, modestZn supplementation improvedZnindices, appeared to induce a cellularFedeficiency and, possibly, further reducedFestatus. In the study by De Brito et al. [353], healthy children of both sexes, aged 8–9 years (*n* = 15), were given a placebo (control group) or 10 mg Zn/day (experimental group) for 3 months. The researchers showed that the decrease in serum Fe was likely due to the effects of chronic Zn administration. The supplementation did not influence levels of Hgb, mean corpuscular volume, ferritin, transferrin, transferrin saturation, ceruloplasmin, or total protein. Similarly, in a study by Antunes et al. [354] on the acute and chronic effects of Zn on the serum Fe profile of children aged 6–9 years, there was a decrease in Fe levels due to Zn supplementation, but no negative effect of Zn on hematocrit and Hgb levels. Wieringa et al. [355] confirmed the positive effect of Fe and Zn supplementation in the prevention of anemia and Zn deficiency.

Holmes et al. [356] studied the effects of multiple micronutrient (MMN) supplementation (15 mg Zn, 65 mg Se, 2 mg Cu) with or without Fe on serum Zn, Se, and Cu concentrations in women from Cambodia. Predominantly anemic nonpregnant women (aged 18–45) received daily 60 mg of Fe (I group); MMN, but no Fe (II group); 60 mg Fe plus MMN (III group); or a placebo (IV group). It was found that 60 mg Fe and the daily MMN formulation may be interfering with the absorption and/or metabolism of supplemental Zn. However, patients took the supplement together with food which was not controlled, and so it is possible that Zn competed for metabolic transporters not only with Fe and minerals contained in the MMN complex but also with elements from the diet. Troost et al. [331] noted that oral Fe therapy may impair Zn absorption. A significant decrease in Zn absorption was observed with 100 and 400 mg of Fe administered in aqueous solution. Unfortunately, that study was limited to a very small group of patients (*n* = 11).

Nguyen et al. [357] conducted a randomized control study on women (*n* = 459) from Guatemala where they received for 12 weeks: (i) weekly 120 mg Fe with 30 mg Zn, (ii) weekly 120 mg Fe, (iii) daily 60 mg Fe with 15 mg Zn, and (iv) daily 60 mg Fe. The combined Fe-Zn supplementation was as effective as Fe alone in improving Fe status, but not effective in Zn status. This may be because Zn and Fe compete with each other only when given in aqueous solution, and if supplementation takes place in the form of a solid meal, such an effect is not recorded, although studies are inconclusive in this case [241,358].

## 7. Cu vs. Zn

The high concentration of Zn in the body caused by the supplementation of this element leads to an increase in the production of metallothionein in the intestinal cells [257]. Cu accumulates in enterocytes due to its high affinity for metallothionein, displacing Zn [359]. As a result of exfoliation of enterocytes in the gastrointestinal tract, Cu concentration is reduced, which may lead to Cu deficiency [360]. A study by Sutton et al. [361] showed that cytopenia in patients can be caused by a high Zn concentration in the body combined with Cu deficiency, and Cu supplementation leads to a reversal of hematologic levels. According to Prasad et al. [362], increased Zn concentration may be caused by improperly fitting dentures, resulting in increased use of dental adhesive, which contains high levels of Zn. Excess Cu can lead to both anemia and neutropenia as a result of the mechanisms in which Cu participates.

Deur et al. [363] concluded that in the case of anemia, Cu deficiency may affect Fe metabolism, including inhibition of Fe absorption from the gastrointestinal tract and shortening of red blood cell life. In the case of neutropenia, on the other hand, an abnormal synthesis of progenitor cells and abnormal maturation and release of neutrophils into the bloodstream occur. Moreover, the removal of granulocytes containing a low Cu concentration is increased [364]. Zn concentration in the body influences SOD activity and Cu level [365]. Extracellular superoxide dismutase and Cu/Zn superoxide dismutase react differently to Zn levels in the body. The extracellular SOD activity increases in response to an increase in Zn levels, as opposed to SOD Cu/Zn whose activity decreases [366].

Zn is used to treat Wilson’s disease, a condition that leads to an excessive accumulation of Cu in the liver [367,368], as Zn supplementation may reduce Cu concentration in the body [344]. Although Zn therapy is often insufficient, it plays an important role as a second-line treatment, supplementing the use of trientine or penicillamine [369]. Sometimes, Zn monotherapy is effective and has therapeutic effects, especially in the neurological Wilson’s disease [370]. Malik et al. [371] reported positive effects of Zn supplementation during pregnancy in WD patients. Optimized doses of Zn have helped to carry out the pregnancy. However, the authors emphasize that pregnancy in these patients has a high risk of preeclampsia. In the case of acrodermatitis enteropathica (AE) patients in Sandström et al. [241] the status of Zn and Cu was measured in response to 1000 and 525 μM/d supplementation. High doses of Zn supplementation were found to decrease Cu absorption and during the dose correction the concentration of Cu was normalised. The authors argue that the primary lesion in AE is a cellular defect in Zn metabolism and not a Zn absorption disorder. Moreover, they point out that in AE it is important to control Zn and Cu concentration due to teratogenic effects of Cu deficiency or excess.

The antagonistic action of Zn and Cu has been the subject of many studies. Wu et al. [372] investigated the effects of dietary Cu and Zn on apparent mineral retention and serum biochemical indicators in young male mink on a corn-fishmeal based diet. It was observed that moderately high Cu in the diet increased Cu retention, but did not reduce Zn absorption, while moderately high Zn in the diet reduced plasma Cu levels.

Eckert et al. [373] showed that the concentration of Zn in the liver does not change after the administration of Cu in the diet, which suggests no interaction between these elements. However, Du et al. [374] observed that the relationship between Zn and Cu depends on the form of Cu administration. Complexes of Cu with protein or lysine caused an increase in the liver concentration of Zn compared to the administration of Cu in the form of inorganic salts, which did not induce any changes in Zn levels.

Zetzsche et al. [375] determine the effect ofhighdietaryZn oxide on trace elementaccumulationin various organs in pigs. They found that dietary Zn supplementation led to Cu co-accumulationin thekidneysofthe pigs.

A study by Baecker et al. [376] in an animal model investigated the effect of high Cu concentration and Zn deficiency on the formation of autism spectrum disorder (ASD). Cu supplementation of pregnant female rats caused a significant decrease in Zn in the brain, above all in the hippocampus. Additionally, the antagonistic action of Cu against Zn caused abnormalities in nerve synapses. Those results suggest that Cu and Zn homeostasis disturbances in pregnant females may contribute to brain underdevelopment and nerve impulse transmission disorders, which may lead to ASD development. Similar conclusions were reached by Reinstein et al. [377] showing an increased risk of fetal malformations with insufficient supply of Zn in the diet and with an increase in Cu. Moreover, Kinnamon [378] reported that Cu and Zn are competing with each other in the fetus and placenta. The result of this competition is a greater absorption of Cu, which is pushing out Zn. Although, Garg et al. [379] in a study on pregnant women who were supplemented with Zn did not report hypocupremia. However, the ratio of both elements should be optimized in pregnancy, to improve reproduction results and reduce the chance of spontaneous abortion [16].

## 8. Conclusions

The presented literature review suggests that Fe, Cu, and Zn are crucial for the proper course of pregnancy. The results should be approached with caution, but most studies indicate the influence of metals on the parameters of mother and child. Moreover, Fe, Cu, and Zn may be promising biomarkers in predicting complications in pregnancy.

Additionally, the results of the researchers show important relationships between Fe, Cu, and Zn in the body. An increase or decrease of one element may significantly affect the action of the other two. It is particularly significant to note that the elements do not exhibit antagonistic actions against each other when they are within daily reference values.

Fe, Cu, and Zn play a key role in the homeostasis of the body, and any changes in their concentrations can cause interactions that are dangerous to the health of the mother and fetus.

## Figures and Tables

**Figure 1 biomolecules-10-01176-f001:**
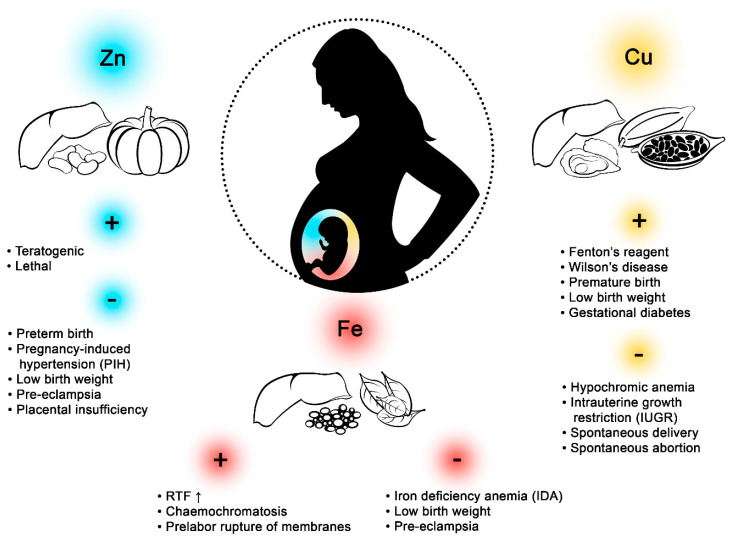
Effects of zinc (Zn), copper (Cu), and iron (Fe) on pregnant women. Excess Zn in the body during pregnancy can have teratogenic or lethal effects. A deficiency of this element may cause preterm birth, pregnancy-induced hypertension (PIH), low birth weight, preeclampsia, and placental insufficiency. High concentrations of Fe in pregnant women may increase the amount of reactive oxygen species (ROS,) lead to hemochromatosis and prelabor rupture of membranes. Fe deficiency, in turn, may lead to iron deficiency anaemia (IDA), low birth weight, and preeclampsia. Hypercupremia may lead to Wilson’s disease and Fenton’s reagent, and may contribute to premature birth, low birth weight, and gestational diabetes. Hypocupremia may lead to hypochromic anemia, intrauterine growth restriction (IUGR), spontaneous delivery, and spontaneous abortion. Large amounts of inorganic Fecan be found in lentils and spinach, while organic Fe is abundant in pork liver. Oysters, beef liver, and cocoa are rich sources of Cu, while calf liver, pumpkin, pork liver, and white beans have high concentrations of Zn.

**Table 1 biomolecules-10-01176-t001:** The concentrations of iron (Fe), copper (Cu), and zinc (Zn) in different biological materials collected from women during childbirth (*n*, number of women; FM, fetal membrane; FS, fetal serum; C, cord; CB, cord blood; CS, cord serum; CP, cord plasma MB, maternal blood; MS, maternal serum; MP, maternal plasma; P, placenta; dw, dry weight; ww, wet weight).

Research Area	*n*	Mean Age(Years)	Concentration of			References
			Fe	Cu	Zn	
**Europe**						
Szczecin, Poland	170	29	P: 640.726 mg/kg dw	P: 6.013 mg/kg dw	P: 66.904 g/kg dw	[2]
			FM: 640.726 mg/kg dw	FM: 8.906 mg/kg dw	FM: 62.788 g/kg dw	
			C: 567.285 mg/kg dw	C: 4.320 mg/kg dw	C: 54.653 g/kg dw	
Poznan, Poland	64	28.1	MS: 1.08 μg/mL	MS: 0.63 μg/mL	MS: 1.91 μg/mL	[3]
			CS (vein): 1.96 μg/mL	CS (vein): 0.65 μg/mL	CS (vein): 0.36 μg/mL	
			CS (artery): 1.63 μg/mL	CS (artery): 0.65 μg/mL	CS (artery): 0.36 μg/mL	
Oleśnica, Poland	64	27	-	MP: 1.93 mg/L	MP: 0.58 mg/L	[4]
				CP: 0.49 mg/L	CP: 0.82 mg/L	
Sverdlovsk region (non-industrial areas) andYekaterinburg (industrial city), Russian	Pregnant women from non-industrial areas (29)	Age range: 17–42	-	MS: 5.44 mkg/mL P: 6.1 mkg/mL	-	[5]
	Pregnant women from industrial city (127)		-	MS: 4.73 mkg/mL P: 13.34 mkg/mL	-	
Moscow, Russian	(150) control	33.1	MS: 1.34 µg/L	MS: 1.15 µg/L	-	[6]
	Pregnancy (169)	33.4	MS: 1.27 µg/L	MS:1.60 µg/L	-	
	Miscarriage (75)	34.8	MS: 1.43 µg/L	MS: 1.12 µg/L	-	
	Infertility (91)	35.5	MS: 1.29 µg/L	MS:1.04 µg/L	-	
Barcelona, Spain	Appropriate for gestational age (96)	32	-	-	MS: 1181 µg /dL ww FS: 1518 µg /dL ww P: 8.4 µg /dL ww	[7]
	Intrauterine growth restriction (49)	32	-	-	MS: 935 µg/dL ww FS: 935 µg/dL ww P: 8.5 µg/dL ww	
	Small for gestational age (33)	30	-	-	MS: 984 µg/dL ww FS: 1134 µg/dL ww P: 8.9 µg/dL ww	
**Africa**						
Mid-Western Region, Nigeria	22	-	P: 84.3 µg/gm dw	P: 6.3 µg/μm dw	P: 66.6 µg/μm dw	[8]
**North America**						
New Hampshire, USA	1159	31.2	-	-	P: 10.26 µg/g ww	[9]
Las Vegas, Nevada, USA	28	29.9	P: 1185.18 µg/L dw	P: 7.81 µg/L dw	P: 63.59 µg/L dw	[10]
New York, USA	Women carrying multiples (101)	30	P: 26.63 µg/g dw	-	P: 5.9 µg/g dw	[11]
	Women carrying singletons (132)	17.4	P: 17.99 µg/g dw	-	P: 2.6 µg/g dw	
Chattanooga, USA	374	-	P: 503,200 µg/kg dw	P: 3889 µg/kg dw	P: 55,120 µg/kg dw	[12]
Pune, India	Normal pregnancies (47)	Age range: 19–35	MB: 120.4 µg/dL	MB: 1.44 µg/dL	MB: 57.5 µg/dL	[13]
			CB: 153.4 µg/dL	CB: 0.26 µg/dL	CB: 90.8 µg/dL	
	Preeclamptic pregnancies (14)		MB: 96.3 µg/dL	MB: 1.58 µg/dL	MB: 49.2 µg/dL	
			CB: 118.6 µg/dL	CB: 0.81 µg/dL	CB:79.9 µg/dL	
Delhi, India	Females delivered full term babies (gestational age > 37 weeks) (50)	25.54	P: 58.94 µg/dL dw	P: 0.255 µg/dL dw	P: 18.28 µg/dL dw	[14]
	Females delivered pre-term babies (gestational age <37 weeks)(30)	24.63	P: 50.60 µg/dL dw	P: 0.220 µg/dL dw	P: 17.26 µg/dL dw	
Jaipur, India	Pregnant women (80)	29.8	P: 72.7 µg/dL	P: 187.3 µg/dL	P: 70.5 µg/dL	[15]
	Non-pregnant women (20)	20	P: 106.9 µg/dL	P: 127.7 µg/dL	P: 100.9 µg/dL	
Ahmedabad, India	Women with spontaneous abortion(159)	24.85	-	MS: 1.59 mg/L	MS: 1.43 mg/L	[16]
	Woman without spontaneous abortion (118)	23.65	-	MS: 1.81 mg/L	MS: 1.46 mg/L	
Hyderabad, India	Pregnant women from rural area (30)	21.1	MS: 201.6μg/dL	MS: 166.6μg/dL	MS: 205.1μg/dL	[17]
			CS: 279.8 μg/dL	CS: 92.03μg/dL	CS: 128.2 μg/dL	
			P: 1159.5 μg/g	P: 78.4 μg/g	P: 49.6 μg/g	
	Pregnant women from urban area (30)	22.2	MS: 128.6 μg/dL	MS: 150.7μg/dL	MS: 245.6μg/dL	
			CS: 200.3 μg/dL	CS:83.5μg/dL	CS: 122.1μg/dL	
			P: 1458.2 μg/g	P: 61.4μg/g	P: 51.5 μg/g	
Jakarta, Indonesia	Pregnant womens ≥37 weeks of gestational age for the term group (25)	27.68	P: 252.16 µg/g dw	P: 2.96 µg/g dw	P: 58.34 µg/g dw	[18]
			CB: 212.00 µg/dL dw	CB: 32.20 µg/dL dw	CB: 293.80 µg/dL dw	
			MS: 77 µg/dL dw	MS: 222.65 µg/dL dw	MS: 45.16 µg/dL dw	
	Pregnant womens preterm birth in 26–36 weeks of gestational age (26)	24.0	P: 78.45 µg/g dw	P: 1.62 µg/g dw	P: 28.41 µg/g dw	
			CB: 236.50 µg/dL dw	CB: 20.60 µg/dL dw	CB: 321.43 µg/dL dw	
			MS: 71.50 µg/dL dw	MS: 215.35 µg/dL dw	MS: 40.26 µg/dL dw	
Fukuoka, Japan	48	29.3	-	P: 3910 ng/g dw	P: 48,100 ng/g dw	[19]
				C: 2960 ng/g dw	C: 35,700 ng/g dw	
Shanghai, China	1568	26.4	MS: 8.1 mmol/L	MS: 23.43 µmol/L	MS: 87.32 µmol/L	[20]
Amman, Jordan	92	27	CB: 116 µg/dL	CB: 49 µg/dL	CB: 114 µg/dL	[21]

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
