# Peer review of "The Role of Fe, Zn, and Cu in Pregnancy"

_biomolecules, 2020, doi:10.3390/biom10081176_

Round 1

Reviewer 1 Report

This review deals with the role of Fe, Cu and Zn in pregnancy. The matter is interesting, although not new. The literature cited is comprehensive. Language is proper.

I have the following major concerns:

  • The general organization of the article is confusing, in that the specific aspects concerning pregnancy are very often reported within the discussion of general aspects concerning each essential micronutrient (absorption, metabolism, related diseases and so on). Consequently, it is difficult for the reader to disentangle the issues of interest for obstetricians from the general issues related to Fe, Cu and Zn. 
  • There is a lack of information on what happens at the placental level: how Fe, Cu and Zn reach the fetus? Is it possible to add some information an this point?
  • Is there any information on the mechanisms leading to preterm birth when there is excess/deficiency of Fe, Cu or Zn in the mother? 
  • Are there meta-analyses on Fe, Cu or Zn in pregnancy worth to be quoted? If yes, they should be stressed in the article

Author Response

Thank you for your review of our paper. We have answered each of your point below.

Reviewer 1

Thank you for your review of our paper. We have answered each of your point below.

This review deals with the role of Fe, Cu and Zn in pregnancy. The matter is interesting, although not new. The literature cited is comprehensive. Language is proper.

I have the following major concerns:

The general organization of the article is confusing, in that the specific aspects concerning pregnancy are very often reported within the discussion of general aspects concerning each essential micronutrient (absorption, metabolism, related diseases and so on). Consequently, it is difficult for the reader to disentangle the issues of interest for obstetricians from the general issues related to Fe, Cu and Zn. 

Thank you very much for the suggestion, but we would like to stay with the current structure of the article because it would be difficult to separate general information from that relating specifically to pregnancy

There is a lack of information on what happens at the placental level: how Fe, Cu and Zn reach the fetus?   to add some information an this point? Corrected

Is there any information on the mechanisms leading to preterm birth when there is excess/deficiency of Fe, Cu or Zn in the mother? Corrected

Are there meta-analyses on Fe, Cu or Zn in pregnancy worth to be quoted? If yes, they should be stressed in the article. Corrected

Reviewer 2 Report

Dear Authors

Your article is well-documented and interesting, especially by the interaction of the different microelements. Please find below my questions. There are also a few typing errors

Paragraph of 2. Iron

The usual plasma concentration of Iron during the pregnancy is not specified (in contrast to the usual plasma concentration given in paragraph 4 for the zinc. Moreover, it was said a few years ago that in the third trimester of pregnancy, women had physiological anemia. Is this still right? Could you precise in this article the physiological values of Fe during the second and third trimester?

Paragraph 3  Copper

Line 206 and 234: typing error   It is 1000 micrograms or 1 mg /day and not 1000 mg/day (which is huge!)

Paragraph 4 Zinc

Line256; could you precise the unit of the Zinc plasma concentration? (microM per…. )

Line362; It could be interesting to specify what means high doses

Paragraph 7  

Line 460: Is the meaning of Cu excess right?  Indeed in the sentence before, it was spoken of Zn excess

Best regards

Author Response

Reviewer 2

Thank you for your review of our paper. We have answered each of your point below.

Your article is well-documented and interesting, especially by the interaction of the different microelements. Please find below my questions. There are also a few typing errors

Paragraph of 2. Iron

The usual plasma concentration of Iron during the pregnancy is not specified (in contrast to the usual plasma concentration given in paragraph 4 for the zinc. Moreover, it was said a few years ago that in the third trimester of pregnancy, women had physiological anemia. Is this still right? Could you precise in this article the physiological values of Fe during the second and third trimester? Corrected

 Paragraph 3  Copper

Line 206 and 234: typing error   It is 1000 micrograms or 1 mg /day and not 1000 mg/day (which is huge!) Corrected

 Paragraph 4 Zinc

Line256; could you precise the unit of the Zinc plasma concentration? (microM per….) Corrected

Line362; It could be interesting to specify what means high doses – Corrected

Paragraph 7  

Line 460: Is the meaning of Cu excess right?  Indeed in the sentence before, it was spoken of Zn excess Corrected

Reviewer 3 Report

“The role of Fe, Zn and Cu in pregnancy” is a highly important topic since these metals are interactive and important for birth outcome. However, this paper may require intensive revision logically. The former part of this manuscript summarized clinical reports and the latter part focused on the biological interaction of minerals. In the latter part it seems that authors did not fully describe the interactions of minerals in “Pregnancy period”. Due to this this manuscript does not look like a single publication.

Also, authors introduced some molecular features/interactions between minerals. I would like to suggest to add information for the molecular absorption, distribution, metabolism and/or excretion of 3 minerals. In addition, absorption and interactions in the intestine and placenta of 3 minerals should be well summarized.

Moreover, I wish authors are able to providing information for the genetic disorders and other minerals. For example, how iron and copper absorbed in the intestine and transported in the placenta in acrodermatitis enteropathica patient who suffers genetical Zn deficiency. Authors may consider MD, WD, hemochromatosis and thalassemia.

  1. In figure 1, authors should clearly demonstrate either pathologies or pathological consequences. Now they are mixed, therefore it is somewhat ambiguous. Zn – should include acrodermatitis enteropathica. Cu – should include MD and spontaneous delivery and absorption are not clear information since authors did not explain fully. Fe may cover hemochromatosis and thalassemia.

  1. [L94] bivalent → divalent

  1. The role of HCP1 in iron metabolism may not crucial since patients with mutations in the gene encoding HCP1/PCFT have folate deficiency, but iron metabolism does not seem to be affected (doi: 10.1002/cphy.c170045.). Therefore, authors should also describe HCP1 is more important for folate acquisition.

  1. [L119] Authors may include increased iron demand such as rapidly growing period and pregnancy for latter description of pregnancy.

  1. [L156] Please check the sentence.

  1. [L203] As I know, the greatest source of copper is oyster.

  1. [L229-231 and 307-308] Please write in details. This paragraph is too short.

  1. [L369] Please consider to add Ha et al.'s previous report (doi: 10.1152/ajpgi.00169.2017.) since this manuscript exhibited high iron consumption decreased copper distribution by using 59Fe.

  1. [L390-392] Please add WD.

Author Response

Thank you for your review of our paper. We have answered each of your point below (green color).

The role of Fe, Zn and Cu in pregnancy” is a highly important topic since these metals are interactive and important for birth outcome. However, this paper may require intensive revision logically. The former part of this manuscript summarized clinical reports and the latter part focused on the biological interaction of minerals. In the latter part it seems that authors did not fully describe the interactions of minerals in “Pregnancy period”. Due to this this manuscript does not look like a single publication.

Also, authors introduced some molecular features/interactions between minerals. I would like to suggest to add information for the molecular absorption, distribution, metabolism and/or excretion of 3 minerals. In addition, absorption and interactions in the intestine and placenta of 3 minerals should be well summarized.

Corrected

Moreover, I wish authors are able to providing information for the genetic disorders and other minerals. For example, how iron and copper absorbed in the intestine and transported in the placenta in acrodermatitis enteropathica patient who suffers genetical Zn deficiency. Authors may consider MD, WD, hemochromatosis and thalassemia.

Corrected

In figure 1, authors should clearly demonstrate either pathologies or pathological consequences. Now they are mixed, therefore it is somewhat ambiguous. Zn – should include acrodermatitis enteropathica. Cu – should include MD and spontaneous delivery and absorption are not clear information since authors did not explain fully. Fe may cover hemochromatosis and thalassemia.

Corrected

 [L94] bivalent → divalent

Corrected  

The role of HCP1 in iron metabolism may not crucial since patients with mutations in the gene encoding HCP1/PCFT have folate deficiency, but iron metabolism does not seem to be affected (doi: 10.1002/cphy.c170045.). Therefore, authors should also describe HCP1 is more important for folate acquisition.

Corrected

[L119] Authors may include increased iron demand such as rapidly growing period and pregnancy for latter description of pregnancy.

Corrected  

[L156] Please check the sentence.

Corrected  

[L203] As I know, the greatest source of copper is oyster.

Corrected

[L229-231 and 307-308] Please write in details. This paragraph is too short.

Corrected

[L369] Please consider to add Ha et al.'s previous report (doi: 10.1152/ajpgi.00169.2017.) since this manuscript exhibited high iron consumption decreased copper distribution by using 59Fe.

Corrected

[L390-392] Please add WD.

Corrected

Round 2

Reviewer 1 Report

After revision, the article substantially improved

Author Response

Thank you for your review of our paper. We have answered each of your point below.

Thank you for your review of our paper. We have answered each of your point below.

English language and style are fine/minor spell check required 

Corrected

After revision, the article substantially improved

Reviewer 3 Report

Please add metal transporters and effect of other metals.

For example, dmt1 -Zn, Cu, atp7a - Fe, zip14 - Fe, and so on.

10.1093/jn/nxy111
10.1039/c6mt00126b
10.1093/jn/nxx041

Author Response

Thank you for your review of our paper. We have answered each of your point below.

(x) English language and style are fine/minor spell check required 

Corrected

Please add metal transporters and effect of other metals.

For example, dmt1 -Zn, Cu, atp7a - Fe, zip14 - Fe, and so on.

10.1093/jn/nxy111
10.1039/c6mt00126b
10.1093/jn/nxx041

Corrected